# Native and Oxidized Low-Density Lipoproteins Increase the Expression of the LDL Receptor and the LOX-1 Receptor, Respectively, in Arterial Endothelial Cells

**DOI:** 10.3390/cells11020204

**Published:** 2022-01-08

**Authors:** Rusan Catar, Lei Chen, Hongfan Zhao, Dashan Wu, Julian Kamhieh-Milz, Christian Lücht, Daniel Zickler, Alexander W. Krug, Christian G. Ziegler, Henning Morawietz, Janusz Witowski

**Affiliations:** 1Department of Nephrology and Internal Intensive Care Medicine, Charité—Universitätsmedizin Berlin, Corporate Member of Freie Universität Berlin and Humboldt-Universität zu Berlin, 10117 Berlin, Germany; lei.chen1991@gmail.com (L.C.); hongfan.zhao@charite.de (H.Z.); Dashan.wu@charite.de (D.W.); christian.luecht@charite.de (C.L.); daniel.zickler@charite.de (D.Z.); jwitow@ump.edu.pl (J.W.); 2Institute of Transfusion Medicine, Charité Universitätsmedizin Berlin, 10117 Berlin, Germany; julian.milz@charite.de; 3Department of Medicine III, University Hospital Carl Gustav Carus Dresden, 01307 Dresden, Germany; alexander.krug@t-online.de (A.W.K.); Christian.Ziegler@uniklinikum-dresden.de (C.G.Z.); 4Department of Medicine III, Division of Vascular Endothelium and Microcirculation, University Hospital Carl Gustav Carus Dresden, Technische Universität Dresden, Fetscherstr. 74, 01307 Dresden, Germany; Henning.Morawietz@uniklinikum-dresden.de; 5Department of Pathophysiology, Poznan University of Medical Sciences, 61-701 Poznan, Poland

**Keywords:** atherosclerosis, endothelium, LDL, LDL receptor, LOX-1, AP-1, NF-κB

## Abstract

Atherosclerotic artery disease is the major cause of death and an immense burden on healthcare systems worldwide. The formation of atherosclerotic plaques is promoted by high levels of low-density lipoproteins (LDL) in the blood, especially in the oxidized form. Circulating LDL is taken up by conventional and non-classical endothelial cell receptors and deposited in the vessel wall. The exact mechanism of LDL interaction with vascular endothelial cells is not fully understood. Moreover, it appears to depend on the type and location of the vessel affected and the receptor involved. Here, we analyze how native LDL (nLDL) and oxidized LDL (oxLDL) modulate the expression of their receptors—classical LDLR and alternative LOX-1—in endothelial cells derived from human umbilical artery (HUAECs), used as an example of a medium-sized vessel, which is typically affected by atherosclerosis. Exposure of HUAECs to nLDL resulted in moderate nLDL uptake and gradual increase in LDLR, but not LOX-1, expression over 24 h. Conversely, exposure of HUAECs to oxLDL, led to significant accumulation of oxLDL and rapid induction of LOX-1, but not LDLR, within 7 h. These activation processes were associated with phosphorylation of protein kinases ERK1/2 and p38, followed by activation of the transcription factor AP-1 and its binding to the promoters of the respective receptor genes. Both nLDL-induced *LDLR* mRNA expression and oxLDL-induced *LOX-1* mRNA expression were abolished by blocking ERK1/2, p-38 or AP-1. In addition, oxLDL, but not nLDL, was capable of inducing *LOX-1* through the NF-κB-controlled pathway. These observations indicate that in arterial endothelial cells nLDL and oxLDL signal mainly via LDLR and LOX-1 receptors, respectively, and engage ERK1/2 and p38 kinases, and AP-1, as well as NF-κB transcription factors to exert feed-forward regulation and increase the expression of these receptors, which may perpetuate endothelial dysfunction in atherosclerosis.

## 1. Introduction

Low-density lipoprotein (LDL) is one of the key cholesterol-carrying particles in the blood. Cells take up LDL after it is bound and internalized by the specific LDL receptor (LDLR) [1]. The LDL-derived cholesterol is then utilized to maintain the integrity of the cell membrane and to exert a variety of regulatory effects, including those related to endogenous cholesterol synthesis and the LDLR pathway itself. Importantly, epidemiological studies show that elevated concentrations of LDL in the blood are strongly associated with the development of atherosclerosis and the resulting cardiovascular disease. Plaque formation is driven by LDL penetrating the dysfunctional endothelium and accumulating in the arterial wall [2]. This process involves both endocytosis and transcytosis, i.e., the transport of LDL to and across the endothelium [3]. These events are greatly accelerated if LDL undergoes oxidative modification and becomes oxidized LDL (oxLDL) [4].

In addition to the classical LDLR-mediated endocytosis, native and modified LDL can be bound and transported by a number of other cellular LDL receptors. These include both the LDL-receptor family members and the scavenger receptor family members (see [5] for a recent review). Not only do the LDLR-independent pathways a play role in lipid homeostasis, but they also modulate intracellular signaling in cells involved in the pathogenesis of atherosclerosis, such as endothelial cells, smooth muscle cells, macrophages, and platelets. The activity of LDL receptors on endothelial cells is of particular interest, as these cells perform barrier functions and regulate the access of macromolecules to the subendothelial space. In this respect, increasing data accumulate to suggest that activation of certain LDL receptors on endothelial cells contributes to endothelial dysfunction and atherosclerosis [3,5,6].

Lectin-like oxidized low-density lipoprotein receptor-1 (LOX-1) is a 50 kDa transmembrane glycoprotein that acts as the main receptor for oxLDL in endothelial cells [7,8]. It has been demonstrated that antisense *LOX-1* reduces the oxLDL-induced production of monocyte-attracting chemokines by the endothelium [9]. On the other hand, endothelial cell-specific overexpression of *LOX-1* leads to a decrease in the production of nitric oxide, but an increase in the production of reactive oxygen species [10]. Moreover, LOX-1 is implicated in oxLDL-induced apoptosis of endothelial cells [11] and their premature senescence [12]. The contribution of LOX-1 to cardiovascular disease is further supported by the observations in mice that overexpression of LOX-1results in accelerated atherosclerosis [10,13,14], while deletion of LOX-1is associated with reduced plaque formation [15,16].

As atherosclerosis affects medium and large arteries, dysfunction of the endothelial cells that line these arteries is a key initial step in disease development [17]. Due to the relative ease of isolation and purification, many experimental studies of endothelial dysfunction use cells derived from umbilical veins or from microcirculation. However, it is increasingly recognized that endothelial cells exhibit substantial functional heterogeneity across various vascular beds [6,18]. Therefore, it is important to study endothelial cells with regard to their exact location and pathophysiological context. In this respect, it has been observed that gene expression profiles differ between arterial and venous endothelial cells, and also between cells from large vessels and microvessels [19]. In addition, it has been suggested that arterial endothelial cells may be more susceptible to atherogenic stimuli than venous endothelial cells, as exposure to oxLDL causes greater changes in gene expression in arterial cells than in venous endothelial cells [20].

Here, we set out to examine whether and how the expression of key LDL receptors in arterial endothelial cells changes in response to native and oxidized LDL.

## 2. Results

### 2.1. HUAEC Culture

Human umbilical artery endothelial cells (HUAECs) in culture showed a cobblestone appearance at confluence (Figure 1A) and expressed the typical endothelial cell markers PECAM-1, VE-cadherin, E-selectin, and von Willebrand factor (Figure 1B–E).

### 2.2. LDL Oxidation

As expected [21], the exposure of LDL isolates to copper sulfate for 24 h led to LDL oxidation, as confirmed by three different methods (Figure 2). As a result of the change in charge upon oxidation [22], oxLDL moved faster during gel electrophoresis than nLDL or EDTA-LDL, causing a marked shift in the corresponding band (Figure 2C).

### 2.3. Cell Viability and LDL Uptake

Exposure of HUAECs for up to 24 h to either nLDL or oxLDL at concentrations up to 100 µg/mL did not impair cell viability, as assessed by the WST-8 conversion assay (Figure 3A) and did not cause cell apoptosis, as shown by the lack of DNA fragmentation (Figure 3B). However, incubation with 100 µg/mL oxLDL for ≥48 h impaired HUAEC viability (not shown). Therefore, all subsequent experiments were performed with exposure times up to 24 h.

Moreover, HUAECs readily internalized LDL, as visualized by fluorescence microscopy using DiI-labeled particles. Within the 3 h time frame assessed, the uptake of oxLDL appeared more pronounced than that of nLDL (Figure 3C). Interestingly, when HUAECs were exposed to DiI-LDL in the presence of a 100-fold excess of unlabeled oxLDL, the uptake of DiI-nLDL was markedly reduced (Figure 3D), suggesting that there could be competition for binding between nLDL and oxLDL, or that both ligands engaged the same signaling pathways.

### 2.4. Effects of nLDL and oxLDL on the Expression of Their Receptors in HUAECs

Exposure of HUAECs to nLDL resulted in a time- and dose-dependent increase in *LDLR* mRNA expression. This effect became significant after 12–24 h of incubation with nLDL at a dose of 100 µg/mL (Figure 4A,C). Moreover, compared to control cells, the exposure to 100 μg/mL nLDL for 24 h increased the abundance of LDLR protein by approximately 30% (Figure 4E). In contrast, exposure of HUAECs to oxLDL under the same conditions did not change significantly *LDLR* mRNA expression (Figure 4A,C), but reduced the abundance of LDLR protein by approximately 50% (Figure 4E). Conversely, the expression of *LOX-1* mRNA (Figure 4B,D) and LOX-1 protein (Figure 3F) increased significantly in response to oxLDL, but not to nLDL. Exposure of HUAECs to 100 µg/mL oxLDL resulted in a rapid increase in *LOX-1* mRNA (Figure 4B,D) and an approximately 40% increase LOX-1 protein (Figure 4F), which both peaked after 7 h of incubation.

### 2.5. Effects of nLDL and oxLDL on ERK1/2 and p38 Activation in HUAECs

As the extracellular signal-regulated kinase 1/2 (ERK1/2) and p38 kinase pathways have been linked to cell response to LDL [23], we assessed phosphorylation of ERK1/2 (pERK1/2) and p38 kinase proteins in HUAECs treated with nLDL or oxLDL. Indeed, exposure to both nLDL and oxLDL resulted in significant increases in the expression of pERK1/2 (5-fold and 2-fold increase, respectively; Figure 5A) phospho-p38 (3.7-fold and 2.8-fold increase, respectively, Figure 5B).

### 2.6. Effects of nLDL and oxLDL on LDLR and LOX-1 Gene Promoters in HUAECs

Exposure of HUAECs to nLDL, but not to oxLDL, led to an increase in the activity of the full length *LDLR* promoter (Figure 6A). Conversely, oxLDL, but not nLDL, activated the *LOX-1* promoter (Figure 6C). Pre-treatment of HUAECs with specific inhibitors to either ERK1/2 or p38 kinases abolished the stimulatory activity of nLDL on the *LDLR* promoter (Figure 6B) and the stimulatory activity of oxLDL on the *LOX-1* promoter (Figure 6D). As LOX-1 affinity for binding oxLDL may be greater for mildly oxidized LDL rather than for more oxidized LDL [24], we compared the effects of oxLDL generated in response to different doses of CuSO_4_. In contrast to the oxLDL produced by 50 µM CuSO_4_, there was no major induction of the *LOX-1* promoter by oxLDL produced by exposure to10 µM CuSO_4_ (Figure 6E).

### 2.7. Involvement of Transcription Factor AP-1 in nLDL-Induced LDLR Expression and oxLDL-Induced LOX-1 Expression

In silico analysis revealed that the promoters of both the *LDLR* gene and the *LOX-1* gene contained high-affinity binding sites for the transcription factor AP-1. To determine whether AP-1 mediates the effects of nLDL and oxLDL on the *LDLR* and *LOX-1* promoters, respectively, electrophoretic mobility shift assays (EMSAs) were performed using a biotin-labeled consensus oligonucleotide for AP-1 binding that corresponded to positions −1614 to −1637 of the *LDLR* promoter and −99 to −122 of the *LOX-1* promoter. These EMSAs showed that nuclear extracts from cells stimulated with nLDL or oxLDL formed distinct DNA-protein complexes with this oligonucleotide (Figure 7A,B). To confirm the involvement of AP-1 in the regulation of *LDLR* and *LOX-1* expression, HUAECs were pre-incubated for 1 h with SR-11302, a specific AP-1 inhibitor, and then stimulated either with nLDL or oxLDL. This analysis showed that inhibition of AP-1 reduced the expression of both nLDL-induced *LDLR* mRNA and oxLDL-induced *LOX-1* mRNA (Figure 7C,D).

### 2.8. Involvement of Transcription Factor NF-κB in oxLDL-Induced LOX-1 Expression

As nuclear factor κB (NF-κB) has been implicated in endothelial cell activation following oxLDL binding (doi. 10.1155/2013/152786), we looked into its role in regulating *LDLR* and *LOX-1* expression. Using EMSA with an oligonucleotide for NF-κB binding that corresponded to positions −735 to −757 of the *LDLR* promoter, we found no evidence of NF-κB binding to this probe after stimulating HUAECs with either nLDL or oxLDL (Figure 8A). In contrast, EMSA using the NF-κB binding fragment that corresponded to positions −89 to −113 of the *LOX-1* promoter revealed formation of a prominent DNA-protein complex in cells treated with oxLDL, but not with nLDL (Figure 8B). Moreover, activation of the *LOX-1* promoter by oxLDL could be substantially reduced in cells with the NF-κB p65 unit silenced with an appropriate siRNA (Figure 8C).

## 3. Discussion

The present study has revealed that nLDL and oxLDL selectively targeted their respective receptors LDLR and LOX-1 in arterial endothelial cells. The common pathway engaged by both nLDL and oxLDL included the transcription factor AP-1 and mitogen-activated protein kinases ERK1/2 and p38. However, while both nLDL and oxLDL upregulated their receptors, the time courses of *LDLR* and *LOX-1* mRNA induction were somewhat different. Exposure of HUAECs to nLDL resulted in rapid activation of ERK1/2, p38, AP-1, and of the *LDLR* promoter, but a significant increase in *LDLR* mRNA and protein expression did not occur until 12–24 h later. Moreover, all these effects were associated with only a rather moderate LDL uptake. Classically, LDLR expression is subject to feedback regulation by LDL-derived cholesterol and mediated by sterol regulatory element-binding proteins (SREBPs) [2]. These exert a tight control over cellular cholesterol homeostasis. It may be that under our experimental conditions the amount of cellular cholesterol remained at a sufficiently high level for an extended period of time and only decreased later, thus allowing the regulatory mechanisms to increase LDLR expression. Such a scenario would suggest some cooperation between the transcription factors SREBP and AP-1. While SREBPs can interact with few other transcription factors to regulate LDLR expression [25], it has not yet been established whether such an interaction exists for AP-1. In the meantime, it has been demonstrated that LDL-containing immune complexes can up-regulate LDLR expression in macrophages through an AP-1-mediated mechanism that is independent of SREBPs [26]. Interestingly, exposure of HUAECs to oxLDL led to a decrease in the abundance of LDLR protein, possibly through a post-translational mechanism, as the levels of *LDLR* mRNA were unaffected. It remains to be determined whether this could be related to the LOX-1 interfering with LDLR recycling [3].

Similar to nLDL, oxLDL also induced rapid activation of ERK1/2, p38 and AP-1. In this case, however, it led to the activation of the *LOX-1* promoter and a significant increase in the expression of *LOX-1* mRNA and protein within just 3 h. This was in contrast to the delayed effect of nLDL on *LDLR*. Moreover, the uptake of oxLDL by HUAECs was clearly more pronounced than the uptake of nLDL. This is consistent with the observations that oxidative modifications of LDL increase their cellular uptake [4]. In macrophages, this results in the formation of lipid-laden foam cells [2]. In human umbilical vein endothelial cells, binding of oxLDL to LOX-1 triggers the production of reactive oxygen species [26]. In line with previous reports, our findings confirm that oxLDL up-regulate its own receptor in endothelial cells [27]. This can create a vicious cycle leading to increased oxLDL accumulation in the vascular wall. The process can be further accelerated by other pro-atherogenic stimuli, including angiotensin II [28], high glucose [29], shear stress [30], and pro-inflammatory cytokines [31,32,33,34].

The observation that the effects of nLDL and oxLDL are mediated by the same pathway suggests that there may be competition between both ligands. In fact, we found that internalization of nLDL was reduced in the presence of an excess of oxLDL. Nevertheless, there are likely to be other signaling pathways responsible for the more specific induction of LDLR and LOX-1 receptors. For example, it has been suggested that the stimulatory effects of oxLDL on *LOX-1* expression are regulated by transcription factors Oct-1 [34] and NF-κB [35]. In addition to previous reports on the NF-κB activation by oxLDL binding to LOX-1 on human umbilical vein endothelial cells [36], here, we demonstrate that in human umbilical artery endothelial cells, oxLDL can upregulate *LOX-1* expression by engaging NF-κB in parallel to signaling through AP-1. *LOX-1* Moreover, the activation of the NF-κB pathway appeared to link specifically oxLDL and LOX-1, as it did not occur in response to nLDL and it did not affect LDLR. The ultimate levels of LOX-1 on cell surfaces can be further controlled by other mechanisms, including microRNAs, proteolysis, and receptor internalization [37].

Endothelial dysfunction resulting from oxLDL-induced LOX-1 activation can manifest as excessive production of monocyte-recruiting chemokines [9], endothelial cell apoptosis [38] or impaired nitric oxide production [39]. It has also been postulated that LOX-1-mediated oxLDL uptake can interfere with caveolae signaling [3].

Given how unfavorable the consequences of LDL overload are, our observations on the effects of nLDL and oxLDL on the expression of their major receptors on arterial endothelial cells expand our understanding of how these complications can be controlled or prevented. In this respect, the fact that both nLDL and oxLDL utilize similar and cross-reacting signaling mechanisms suggests that the identified kinases (ERK1/2 and p38) and transcription factors (AP-1) may not be sufficiently selective targets for intervention. In contrast, NF-κB signaling in arterial endothelial cells, together with preventing LDL oxidation or directly targeting LOX-1 may prove more effective options.

## 4. Materials and Methods

### 4.1. Materials

Unless specified otherwise, all cell culture reagents were from Invitrogen (Karlsruhe, Germany) and all chemicals were from Sigma–Aldrich (Munich, Germany).

### 4.2. Cell Culture

Human umbilical artery endothelial cells (HUAECs) were isolated from the arteries of normal umbilical cords by digestion with collagenase II [40]. The umbilical cords were collected after normal delivery from consenting patients and processed according to a protocol approved by the Ethics Committee of the Carl Gustav Carus Faculty of Medicine at the Technische Universität Dresden (decision no. EK124082003). Cells were identified by the cobblestone-like appearance at confluence (Figure 1A), by the expression of endothelial cell markers, as assessed by flow cytometry (Figure 1B–E), and by the ability to internalize LDL particles (see below). Flow cytometry was performed as reported previously [41,42,43,44], upon labeling trypsin-detached cells with FITC- or PE-conjugated monoclonal antibodies against human CD31 (PECAM-1), CD62 (E-selectin), CD144 (VE-cadherin), and von Willebrand factor (vWF) or with isotype control antibodies (all from Becton Dickinson, Heidelberg, Germany). To minimize donor-related variability, equal numbers of primary HUAECs from 3 separate donors were pooled and used throughout the study. Cells were maintained in M199 medium supplemented with 10% fetal calf serum and 1% growth supplement (PromoCell, Heidelberg, Germany) [45]. Prior to experiments, confluent HUAECs were incubated in serum-free medium for 5 h.

### 4.3. Lipoprotein Isolation and Oxidation

LDL were isolated from EDTA-collected plasma of healthy donors by NaBr gradient centrifugation and then purified by dialysis with 3 L phosphate-buffered saline (PBS), as previously described [46]. The native LDL (nLDL) obtained was then oxidized with 50 µM CuSO_4_ for 24 h [21] and the oxidation level was estimated by measuring oxidized Apo-B-100 with an immunoassay (Mercodia oxLDL ELISA Kit, Uppsala, Sweden), conjugated diene formation by spectrophotometry at 234 nm, and gel electrophoretic mobility using Hydragel LDL/HDL CHOL Direct K20 kit (Sebia, Norcross, GA, USA). The preparations of nLDL and oxLDL remained stable when stored at 4 °C for up to 4 weeks, as determined by gel electrophoresis and repeated measurements of conjugated diene formation. In the experiments, nLDL and oxLDL were added to culture media at concentrations as specified in figure legends, using PBS as a vehicle.

### 4.4. Assessment of Cell Viability and Apoptosis

Cell viability was assessed by measuring mitochondrial activity using the water-soluble tetrazolium salt (WST-8) conversion assay (PromoCell, Heidelberg, Germany), as per manufacturer’s instructions. Apoptosis of LDL-treated HUAECs was assessed by DNA-fragmentation. To this end, cells were lysed overnight with a buffer consisting of Tris-HCl (10 mM, pH 7.8), EDTA (5 mM, pH 8.0), 0.5% (*w*/*v*) sodium dodecyl-sulfate (SDS), and proteinase K (100 mg/mL). Genomic DNA was then extracted, as described [47], and run on an agarose gel stained with ethidium bromide, visualized by UV transillumination, and photographed.

### 4.5. LDL Uptake

The uptake of LDL by HUAECs was assessed following nLDL/oxLDL labeling with 1,1′-dioctadecyl-3,3,3′,3′-tetramethylindocarbocyanine perchlorate (DiI) (Harbor Bioproducts, Norwood, MA, USA). Confluent HUAECs were incubated in 0.5% (*v*/*v*) FCS-containing medium for 24 h and then exposed to DiI-labeled LDL (100 μg/mL) for 3 h. After that the cells were washed with ice-cold phosphate-buffered saline (PBS, pH 7.4), fixed in 4% paraformaldehyde for 30 min and assessed by fluorescence microscopy.

### 4.6. Gene Expression Analysis

Total RNA from HUAECs was extracted by lysis with guanidinium isothiocyanate and centrifugation through cesium chloride solution [48]. Then, 500 ng of total RNA was reverse transcribed using random hexamer primers and SuperScript II reverse transcriptase (Invitrogen, Karlsruhe, Germany) [49]. Specific primers for PCR were selected by homology search in GenBank as follows: LOX-1 (*LOX-1*; GenBank NM_001172632) forward primer 5′-ACTCTCCATGGTGGTGCCTGG-3′, and reverse primer 5′-CATTCAGCTTCCGAGCAAGGG-3′; LDLR (*LDLR*; GenBank NM_000527.5) forward primer 5′-TGGCATCACCCTAGATCTCC-3′, and reverse primer 5′-CAGCCAACAAGTTGACATCG-3′; 18S rRNA (*RNA18SN5*; and GenBank NR_003286.4) forward primer 5′-GTTGGTGGAGCGATTTGTCTGG-3′, and reverse primer 5′-AGGGCAGGGACTTAATCAACGC-3′. The target cDNA sequences were amplified by PCR using 400 nM specific and antisense primers, 1× Taq reaction buffer, 20 nM of each dNTP, and 1 U rTaqDNA polymerase (Amersham Biosciences Europe, Freiburg, Germany). After initial denaturation at 95 °C for 2 min, PCR amplification was performed using the following protocol: 30 s at 95 °C, 30 s at a specific annealing temperature, and 30 s at 72 °C. PCR was performed for 35 cycles for *LDLR* and *LOX-1*, and for 11 cycles for *RNA18SN5*, as under these conditions the PCR products were generated during the exponential phase of amplification. After final extension at 72 °C for 3 min, PCR products were separated by electrophoresis on standard agarose gels stained with ethidium bromide and documented by GeneFlash gel documentation system (SYNGENE Europe, Cambridge, UK). Optical density of amplified PCR fragments was quantified using AIDA software (Raytest, Berlin, Germany) and normalized to the density of 18S rRNA PCR fragments.

### 4.7. Western Blot Analysis

After exposure to LDL, HUAECs were washed twice with PBS and cell membrane proteins were isolated by three freeze–thaw cycles and centrifugation at 14,000× *g* for 5 min. The protein concentration of the supernatant obtained was determined with the Pierce BCA Protein Assay Reagent (Perbio Science, Bonn, Germany). Equal amounts of protein (50 µg/lane) were separated by SDS-polyacrylamide gel electrophoresis (SDS-PAGE; 10%) and transferred to nitrocellulose membranes (Schleicher & Schuell, Dassel, Germany). Membranes were incubated with primary antibodies against pERK-1/2, phospho-p38-MAPK (Thr180/Tyr182) (Cell Signaling Technology, Frankfurt, Germany), LDLR (R&D Systems/Bio-Techne, Wiesbaden, Germany), and LOX-1 (JTX92; kindly provided by Prof. T. Sawamura; Department of Vascular Physiology, National Cerebral and Cardiovascular Center Research Institute, Osaka, Japan), followed by secondary horseradish peroxidase-linked rabbit IgG (Amersham Biosciences Europe, Freiburg, Germany). Bound proteins were detected with ECL Western blotting detection reagent (Amersham Biosciences Europe, Freiburg, Germany) and quantified by densitometry using AIDA software (Raytest, Berlin, German).

### 4.8. Quantification of Protein Phosphorylation

Quantification of p38 and ERK1/2 phosphorylation was performed using specific immunoassays, essentially as described by Versteeget al. [50].

### 4.9. Promoter Analysis

The construct for the full-length *LOX-1* gene promoter was kindly provided by Prof. T. Sawamura (Department of Vascular Physiology, National Cerebral and Cardiovascular Center Research Institute, Suita, Osaka, Japan). The *LDLR* full-length promoter fragment was generated by PCR amplification using genomic DNA from HUEACs isolated with Isol-RNA Lysis Reagent solution (5Prime, Hamburg, Germany) and the appropriate primers. The Infusion Cloning Kit (Clontech; Takara Bio USA, Mountain View, CA, USA) was used together with the pGL4.10 vector backbone to create the luciferase reporter constructs. The correct length of the promoter segments was checked by restriction digest.

### 4.10. Transient Transfections and Luciferase Assays

For transient transfection studies, HUAECs were seeded into 6-well culture plates and allowed to reach 70–80% confluence. Transfections were performed in the absence of serum using the TurboFect™ transfection reagent (ThermoFisher Scientific, Darmstadt, Germany), according to the manufacturer’s instructions. HUAECs were transfected with the *LOX-1* or *LDLR* reporter plasmids and the reference plasmids, and assayed with the dual-luciferase reporter assay system (Promega, Mannheim, Germany) as described previously [51]. Transfections with siRNAs were performed with the siRNA Transfection Reagent and siRNAs for NFκB p65 (sc-29410) or with scrambled siRNA control (sc-37007) as per the manufacturer’s instructions (all materials from Santa Cruz Biotechnology). Luciferase activity was expressed as relative light units.

### 4.11. Computational Analysis of the LOX-1 and LDLR Promoter

The human *LOX-1* promoter region −2040 to −1 (GenBank NC_000012.12) and the *LDLR* promoter region −2074 to −1 (GenBank NC_000019.10) were analyzed with AliBaba2.1 software http://gene-regulation.com/pub/programs/alibaba2/ (accessed on 1 December 2021) for the presence and location of potential transcription factor binding sites.

### 4.12. Nuclear Extracts and Electrophoretic Mobility Shift Assay

Nuclear extracts were prepared using the NE-PER Nuclear and Cytoplasmic Extraction Kit (ThermoFisher Scientific, Darmstadt, Germany) according to the manufacturer’s instructions. The extracts obtained were aliquoted and stored at −80 °C. Oligonucleotide probes were labeled with the Biotin 3′ End DNA Labeling Kit (ThermoFisher Scientific, Darmstadt, Germany). For the electrophoretic mobility shift assay (EMSA), the following probes were used (the corresponding promoter region is given in parentheses): *AP1* 5′-AAGAATTTGCGTCAGCGAACTTCC-3′ (−99 to −122); *NFκB* 5′-CGTCAGCGAACTTCCCAATATGAA-3′ (−89 to −113) for *LOX-1,* and *AP1* 5′-GATCCACCTGCCTCAGCCTCCCAA-3′ (−1614 to −1637); *NFκB* 5′- TTCAACTGTGAAAGCCCTGTTTTG-3′ (−735 to −757) for *LDLR*. Each binding mixture (20 µL) for EMSA contained nuclear extract (5 µg), labeled double-stranded probe (20 fM), poly-dI/dC (1 µg), and 10× reaction buffer (2 µL) and was incubated at room temperature for 30 min. The protein-DNA complexes formed were analyzed by electrophoresis in 6% non-denaturing polyacrylamide gels and visualized using a LightShift Chemiluminescent EMSA Kit (ThermoFisher Scientific, Darmstadt, Germany).

### 4.13. Statistical Analysis

Statistical analysis was performed using GraphPad Prism 9.3.0 software (GraphPad Software, La Jolla, CA, USA). The data were analyzed with the t-test or analysis of variance, followed by the Tukey’s test for multiple comparisons, as appropriate. Results were expressed as the mean ± SD. Differences with a *p* value < 0.05 were considered significant and indicated with asterisks.

## Figures and Tables

**Figure 1 cells-11-00204-f001:**
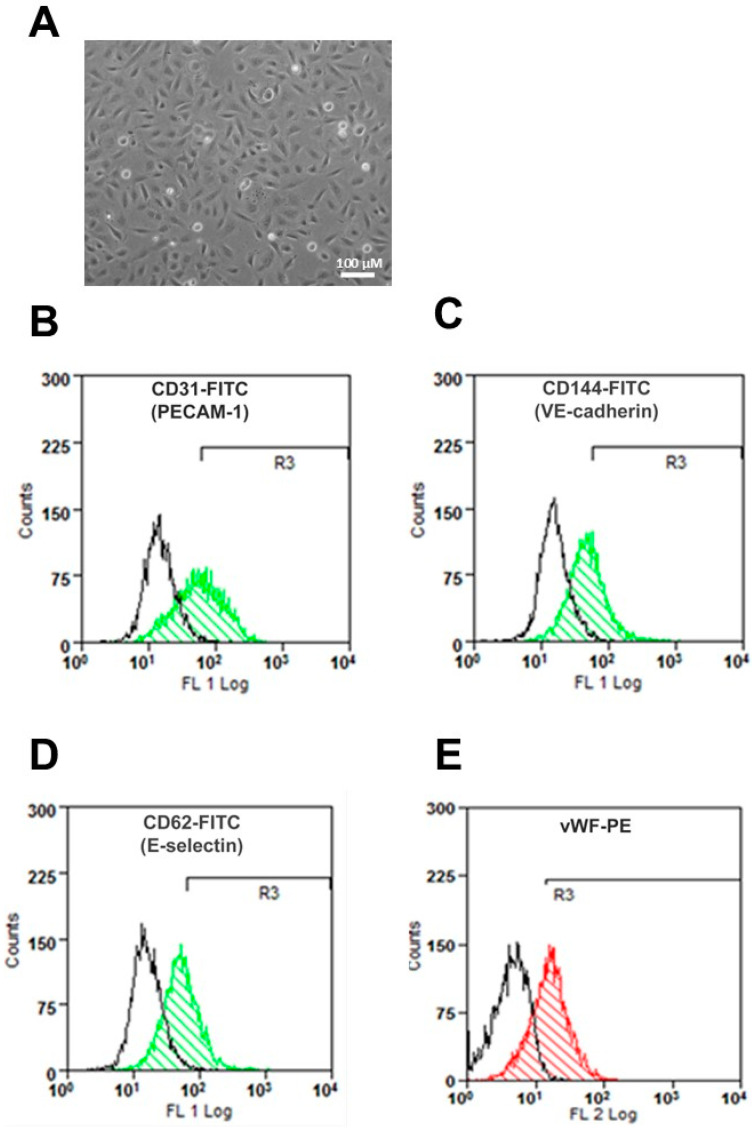
Characterization of human umbilical artery endothelial cells (HUAECs) in culture. (**A**) HUAEC morphology; (**B**–**E**) HUAECs were analyzed by flow cytometry after staining either with antibodies against endothelial cell-specific biomarkers (as indicated in color) or with isotype control IgG (in black). Representative histograms are shown.

**Figure 2 cells-11-00204-f002:**
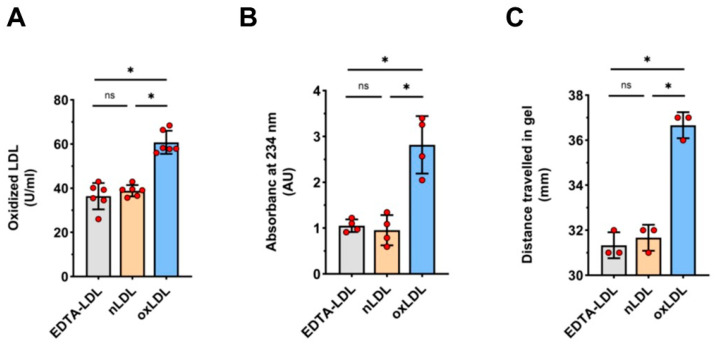
LDL oxidation. Three LDL fractions were assessed: LDL freshly isolated from EDTA-plasma (EDTA-LDL),) LDL further purified by dialysis (nLDL), and nLDL treated with CuSO4 for 24 h (oxLDL). The level of oxidation was measured by (**A**) ELISA (*n* = 6), (**B**) absorbance at 234 nm (*n* = 4), and (**C**) gel electrophoretic mobility (*n* = 3), as described in Methods, with *n* referring to a number of separate LDL preparations. In (**C**), an exemplary gel is shown. t-test mean +/− SD with * *p* < 0.05.

**Figure 3 cells-11-00204-f003:**
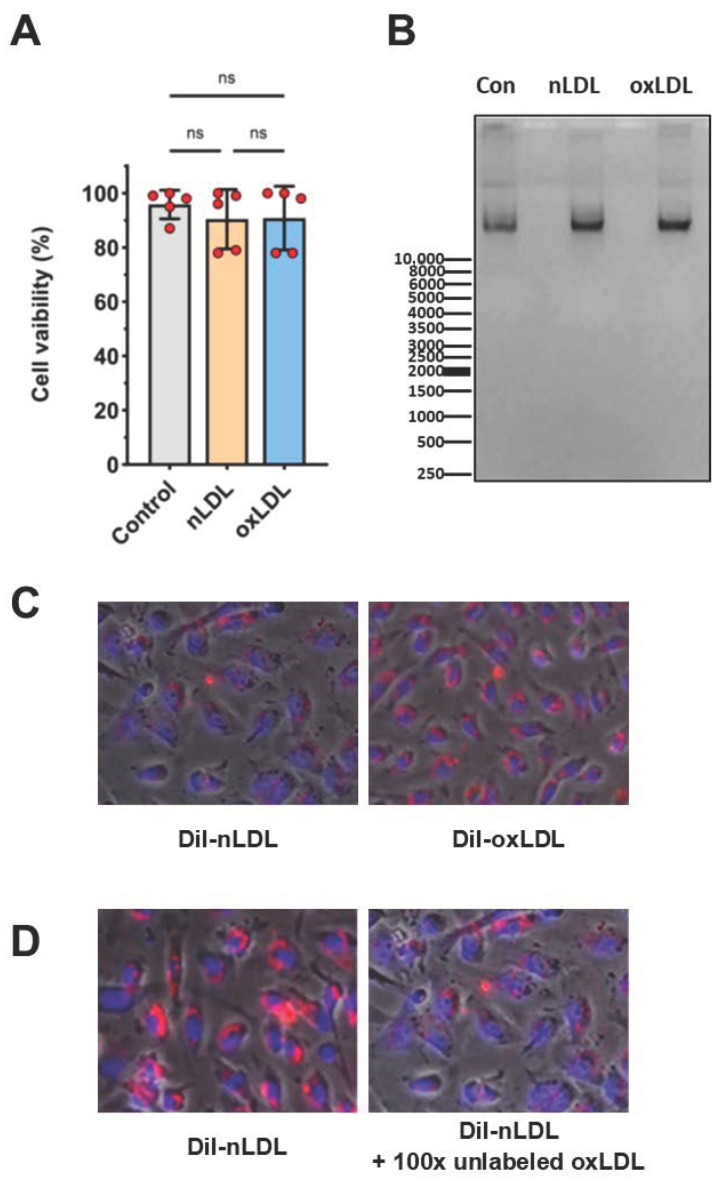
Cell viability and LDL uptake. (**A**) The percentage of viable cells was measured with the WST-8 assay following 16 h of incubation with either nLDL or oxLDL at 100 µg/mL; *n* = 5. (**B**) Cell apoptosis was assessed by DNA-fragmentation following exposure of HUAECs to 100 μg/mL nLDL/oxLDL for 24 h. DNA was separated by electrophoresis on a 0.8% agarose gel at 90 V. (**C**,**D**) Uptake of DiI-labeled LDL by HUAECs was assessed by fluorescence microscopy using 450 nm and 490 nm filters after 3 h of incubation with (**C**) 100 μg/mL of either DiI-nLDL or DiI-oxLDL or with (**D**) 100 μg/mL of DiI-nLDL in the presence or absence of a 100-fold excess of unlabeled oxLDL. Magnification 40×, scale 100 µm.

**Figure 4 cells-11-00204-f004:**
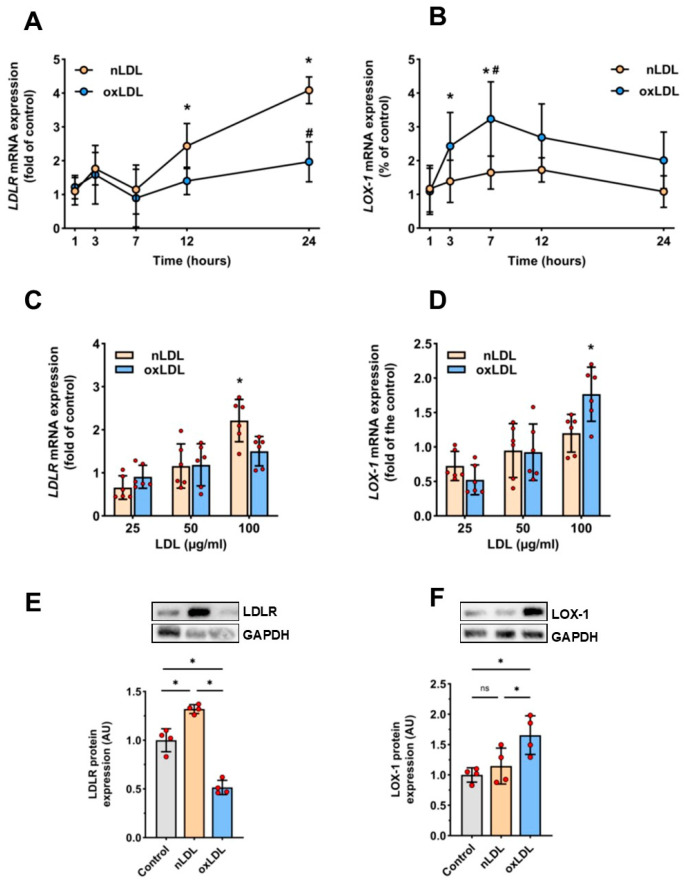
Effect of nLDL and oxLDL on LDLR and LOX-1 expression. (**A**–**D**) Total RNA was harvested from HUAECs treated with either nLDL or oxLDL, and analyzed for *LDLR* and *LOX-1* mRNA expression by reverse transcription-PCR. The results are presented in relation to the expression levels in untreated control cells. In A and B, HUAECs were incubated in the presence or absence of 100 µg/mL of nLDL or oxLDL for the times indicated (*n* = 6). In C and D, HUAECs were incubated for 12 h with either nLDL or oxLDL at doses indicated (*n* = 6). (**E**,**F**): Total protein was extracted from HUAECs incubated with 100 µg/mL of either nLDL or oxLDL for 24 h or (**F**) 7 h, and expression of target proteins was measured by Western blotting. Asterisks and hash signs represent a significant difference compared to control cells and nLDL-treated cells, respectively. t-test mean +/− SD with vs. 1 h * *p* < 0.05 and # vs. nLDL *p* < 0.005 ns = not significant.

**Figure 5 cells-11-00204-f005:**
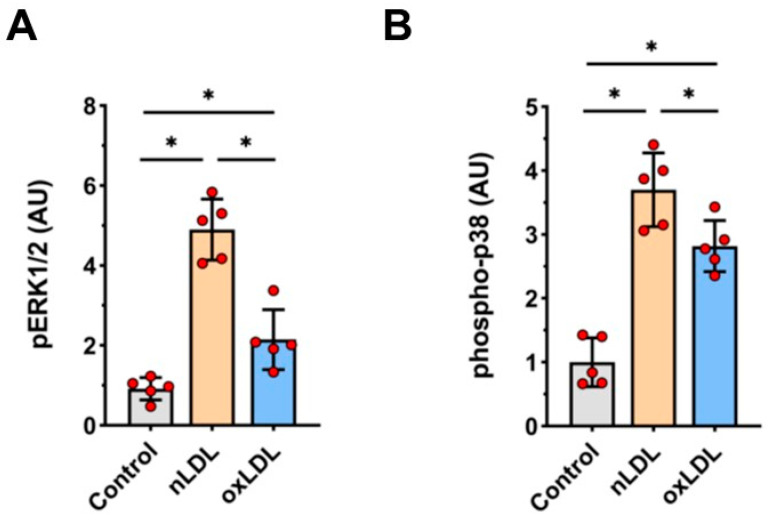
Effect of nLDL and oxLDL on ERK1/2 and p38 phosphorylation. HUAECs were stimulated with 100 μg/mL of either nLDL or oxLDL for 20 min and immediately analyzed by Western blotting for the presence of phosphorylated ERK1/2 (**A**) and p38 (**B**). t-test mean +/− SD with * *p* < 0.05.

**Figure 6 cells-11-00204-f006:**
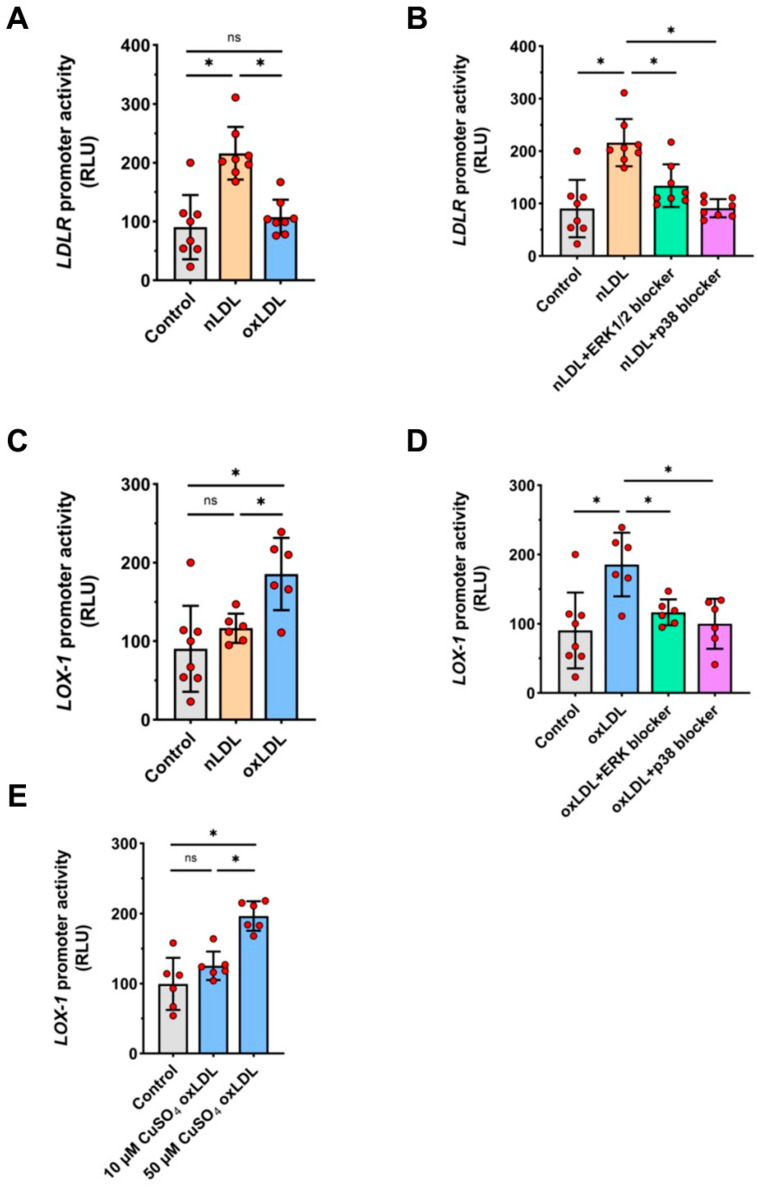
Activation of *LDLR* and *LOX-1* gene promoters by nLDL and oxLDL. HUAECs were transiently transfected with full-length promoter constructs for *LDLR* (**A**,**B**) or *LOX-1* (**C**–**E**) and then stimulated with 100 μg/mL of nLDL or oxLDL for 3 h and analyzed for luciferase activity. In (**B**,**D**), cells were pre-treated for 1 h with 100 nM of PD184352 (ERK1/2 inhibitor), 100 nM of PD-169316 (p38 inhibitor) or vehicle prior to stimulation. In (**E**), cells were treated with the same dose of oxLDL (100 μg/mL), but generated in response to different concentrations of CUSO_4_ (10 µM vs. 50 µM). t-test mean +/− SD with * *p* < 0.05 and ns = not significant.

**Figure 7 cells-11-00204-f007:**
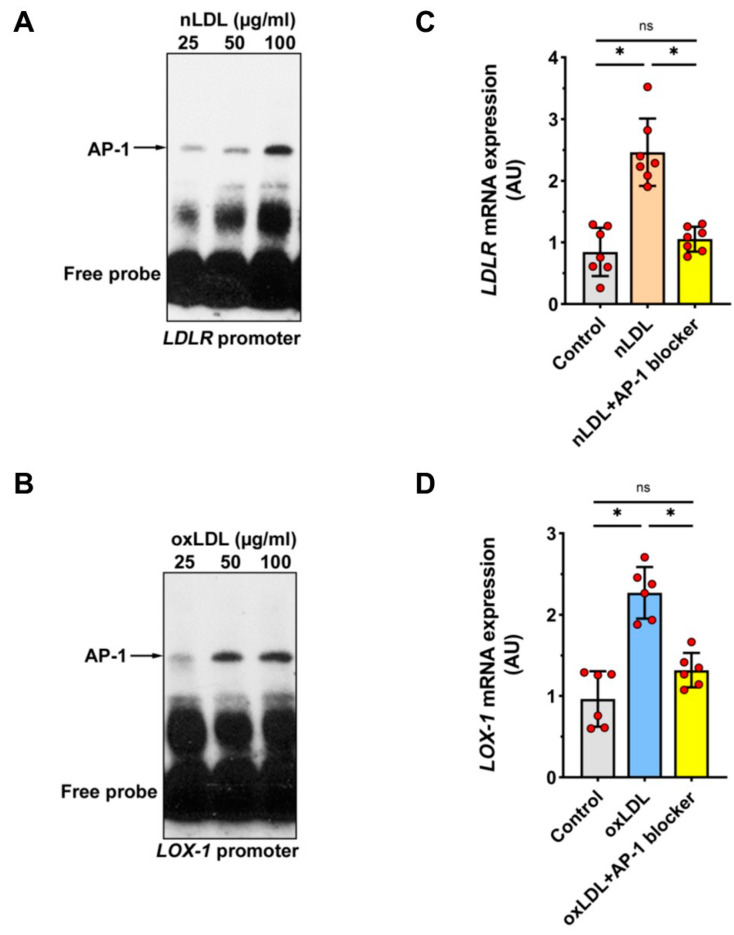
Identifying the role of AP1 in nLDL-induced *LDLR* expression and oxLDL-induced *LOX-1* expression. (**A**,**B**) Nuclear extracts were obtained from HUAECs incubated for 3 h with either nLDL (**A**) or oxLDL (**B**), at doses as indicated, and analyzed by EMSA using consensus oligonucleotides for AP-1 binding. (**C**,**D**) HUAECs were preincubated for 1 h with or without the AP-1 inhibitor SR-11302 (5 µM) and then stimulated with 100 µg/mL of nLDL or oxLDL for 12 h and analyzed for *LDLR* mRNA (**C**) and *LOX-1* mRNA (**D**) expression. t-test mean +/− SD with * *p* < 0.05 and ns = not significant.

**Figure 8 cells-11-00204-f008:**
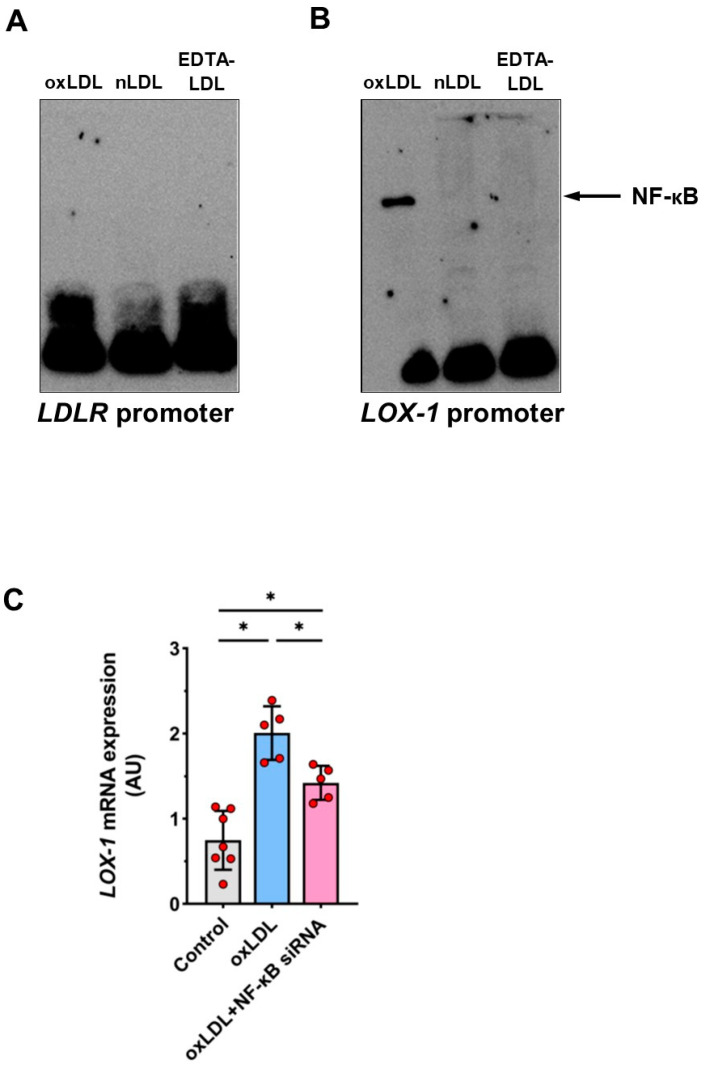
The role of NF-κB in the *LOX-1* promoter activation by oxLDL. (**A**,**B**) Nuclear extracts were obtained from HUAECs incubated for 6 h with EDTA-LDL (control), nLDL or oxLDL (all at 100 µg/mL) and analyzed by EMSA using oligonucleotides for NF-κB binding within the *LDLR* promoter (**A**) or the *LOX-1* promoter (**B**). In (**C**), HUAECs were preincubated with or without siRNA for the subunit p65 of NFκB (10 µM) for 24 h and then stimulated with 100 µg/mL of oxLDL for 12 h and analyzed for *LOX-1* mRNA expression. t-test mean +/− SD with * *p* < 0.05.

## Data Availability

Original data are available upon request.

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
