# Peer review of "Native and Oxidized Low-Density Lipoproteins Increase the Expression of the LDL Receptor and the LOX-1 Receptor, Respectively, in Arterial Endothelial Cells"

_cells, 2022, doi:10.3390/cells11020204_

Round 1

Reviewer 1 Report

In the present manuscript, Catar et al. study the changes in LDL and LOX-1 receptor (mRNA and protein levels) following HUAECs incubation with native LDL (nLDL) and oxidized LDL (oxLDL). The work seems well-conducted, it is well-written, experimental data is properly presented and analyzed, and conclusions are adequately supported by the obtained results.

A main concern would be the novelty of the presented findings, which should be better highlighted throughout the manuscript. In addition, it would be interesting to evaluate at which extent the pathway that the authors propose might further contribute (or not) to nLDL/oxLDL-induced endothelial disfunction. In other words, does LDL/LOX-1 receptor upregulation by nLDL/oxLDL promote larger changes in mRNA/protein levels of endothelial adhesion molecules (e.g. selectins and integrins) when compared to conditions in which LDL/LOX-1 receptor upregulation by nLDL/oxLDL is blocked?

Reviewer 2 Report

This paper analyzes the effect of nLDL and oxLDL on the expression of LDLR and LOX-1 in HUAECs, as well as the intracellular pathways involved. Although the manuscript is overall well-written and the design of the study is proper, I have some concerns about the interpretation and novelty/importance of the results.

The results show that the effect of nLDL and oxLDL are mediated by the same pathways; such observation suggests that there could be a competition between both ligands. In this regard, it would be useful, for example, to assess the internalization of DiI-nLDL in the presence of high doses of unlabeled oxLDL. Because nLDL and oxLDL seem to share the intracellular pathway, the putative factors determining the specific induction of one or other receptor should be further discussed.

The authors focus the study in some specific intracellular pathway molecules (ERK, p38, AP-1), but other feasible candidates are not evaluated (or is not shown in the manuscript), mainly NF-kB.

In the study, 50 mM of copper 24 h are used to oxidize LDL, leading to an extensively oxidized form of LDL. Did the authors check the effect of mildly oxidized LDL (with concentrations of copper (5-10 mM))? It has been described that LOX-1 binds mildly oxidized LDL with higher affinity than extensively oxLDL.

Although it is well-known that oxLDL induces LOX-1 expression in endothelial cells, it has been reported that, in HCAEC. oxLDL at 100 mg/ml decreases LOX-1 expression, due to its cytotoxic effect. However, in the current study, no apoptotic/cytotoxic effect has been found, in spite of using an extensively modified LDL. Owing to the importance of cytotoxicity and inflammation in endothelial cells in the context of atherosclerosis, could the authors discuss these points?

In my opinion, due to the existence of many previous studies about LDL and their receptors in endothelial cells, the novelty of the findings should be highlighted in the Discussion, and the necessity and the aim of the study clearly established in the Introduction.

Other points:

  • In the title is not clear whether both LDLs induce both receptors.
  • The meaning of the sentences in lines 95 and 120-121 is not clear.
  • In Figure 2C the scale bar should be included and the quality of the image improved.
  • In Methods, the incubation conditions of LDLs with cells should be included (buffer in the dialysis of LDL, concentration of LDL and number of cells, culture medium).
  • In the statistical section, the specific tests used should be indicated.

Reviewer 3 Report

Reviewing the manuscript entitled, “Native and oxidized low-density lipoproteins selectively target 2 and increase the expression of LDL and LOX-1 receptors in ar-3 terial endothelial cells” by Catar R et al., this is an article focusing on differences in nLDL and oxLDL receptor expression mechanism in the HUAECs. The authors need to respond to the following concerns to reach an acceptable quality.

Concerns

The author should add an experiment that show how to recognize the difference in signal from LDLR or LOX1 even though the intracellular signals are the same. Then, you should describe and draw how the signal from LDLR increases LDLR and the signal from LOX1 increases LOX1 in figure6 and in the Discussion.

From line 252 to 254, the authors mentioned “To minimize donor-related variability, 252 approximately equal numbers of primary HUAECs from 3 separate donors were pooled and used throughout the study.” If so, ethical review approval is required. The authors should add the ethical statement.

This study requires proof that HUEACs are working properly. The authors should add that.

LOX1 and LOX-1 are mixed. The authors should modify that.

In figure1, why does gel electrophoretic mobility come with only C?

In figure3 E, F, the results of Western blot analysis are low quality. The authors need to modify them.

Round 2

Reviewer 1 Report

The authors have properly addressed my comments. I have no further concerns.

Author Response

We are grateful for the positive evaluation of our revised paper.

Reviewer 2 Report

I think that the manuscript have improved and my concerns have been addressed rather adequately

Author Response

(The authors gave the same response as above.)

Reviewer 3 Report

Reviewing the revised manuscript entitled, “Native and oxidized low-density lipoproteins selectively target 2 and increase the expression of LDL and LOX-1 receptors in ar-3 terial endothelial cells” by Catar R et al., the authors well responded my concerns except the following one. After respond this, this manuscript will reach to acceptable quality.

Concerns

The authors should add “an exemplary gel” in Figure2 A, B. Otherwise, you should delete an exemplary gel in Figure2 C.

Author Response

We are grateful for the positive evaluation of our revised paper and deleted the exemplary gel in Figure 2C as recommended by the reviewer.